

# 1 Contrasting potential for biological N₂-fixation at three

# 2 polluted Central European *Sphagnum* peat bogs: Combining

# 3 the ¹⁵N₂-tracer and natural-abundance isotope approaches

Marketa Stepanova[1], Martin Novak[1*], Bohuslava Cejkova[1], Ivana Jackova[1], Frantisek Buzek[1],
Frantisek Veselovsky[2], Jan Curik[1], Eva Prechova[1], Arnost Komarek[3], Leona Bohdalkova[1]
[1]Department of Environmental Geochemistry and Biogeochemistry, Czech Geological Survey, Geologicka 6,
00 Prague 5, Czech Republic
[2]Department of Rock Geochemistry, Czech Geological Survey, Geologicka 6, 152 00 Prague 5, Czech Republic
[3]Department of Probability and Mathematic Statistics, Faculty of Mathematics and Physics, Charles University,
Sokolovska 83, 186 75 Prague 8, Czech Republic
*Correspondence to*: martin.novak2@geology.cz
**ABSTRACT**
Availability of reactive nitrogen ($N_r$) is a key control of carbon (C) sequestration in wetlands. To complement
the metabolic demands of *Sphagnum* in pristine rain-fed bogs, diazotrophs supply additional $N_r$ *via* biological
nitrogen fixation (BNF). Since breaking the triple bond of atmospheric $N_2$ is energy-intensive, it is reasonable to
assume that increasing inputs of pollutant $N_r$ will lead to BNF downregulation. Yet, recent studies have
documented measurable BNF rates in *Sphagnum*-dominated bogs also in polluted regions, indicating adaptation
of $N_2$-fixers to changing N deposition. Our aim was to quantify BNF at high-elevation peatlands located in
industrialized Central Europe. A ¹⁵N₂-tracer experiment was combined with a natural-abundance N-isotope study
at three *Sphagnum*-dominated peat bogs in the northern Czech in an attempt to assess the roles of individual
BNF drivers. High short-term BNF rates ($8.2 \pm 4.6$ g N m² d⁻¹) were observed at Male Mechove Jezirko
receiving ~17 kg $N_r$ ha⁻¹ yr⁻¹. The remaining two peat bogs, whose recent atmospheric $N_r$ inputs differed from
Male Mechove Jezirko only by 1-2 kg ha⁻¹ yr⁻¹ (Uhlirska and Brumiste), showed zero BNF. The following
parameters were investigated to elucidate the BNF difference: $NH_4^+$-N/$NO_3^-$-N ratio, temperature, wetness,
*Sphagnum* species, organic-N availability, possible P limitation, possible Mo limitation, $SO_4^{2-}$ deposition, and
pH. At Male Mechove Jezirko and Uhlirska, the same moss species (*S. girgensohnii)* was used for the ¹⁵N₂
experiment, and therefore host identity could not explain the difference in BNF at these sites. Temperature and
moisture were also identical in all incubations and could not explain the between-site differences in BNF. The
N:P stoichiometry in peat and bog water indicated that Brumiste may have lacked BNF due to P limitation,
whereas non-detectable BNF at Uhlirska may have been related to 70 times higher $SO_4^{2-}$ concentration in bog
water. Across the sites, the mean natural-abundance $\delta^{15}N$ values increased in the order: atmospheric deposition (-



5.3 ± 0.3 ‰) < *Sphagnum* (-4.3 ± 0.1 ‰) < bog water (-3.9 ± 0.4 ‰) < atmospheric $N_2$ (0.0 ‰). Only at
Brumiste, N in *Sphagnum* was significantly isotopically heavier than in atmospheric deposition, possibly
indicating a longer-term BNF effect. Collectively, our data highlight spatial heterogeneity in BNF rates under
high $N_r$ inputs and the importance of environmental parameters other than atmospheric $N_r$ pollution in regulating
BNF.
*Keywords*: Peat, *Sphagnum*, nitrogen deposition, pollution, biological nitrogen fixation, BNF controls,
phosphorus limitation
**1. Introduction**
Nitrogen (N) is the limiting nutrient in most terrestrial environments. The amount and form of N available to
organisms (reactive N, $N_r$) is controlled by biogeochemical processes (Vitousek and Howarth, 1991; LeBauer
and Treseder, 2008; Zhang et al., 2020; Davies-Barnard and Friedlingstein, 2020). A growing body of research
has focused on the role of biological $N_2$-fixation (BNF) as a source of $N_r$ in pristine ecosystems, such as
subarctic tundra and boreal forests, with special attention given to ombrotrophic peat bogs and minerotrophic
fens (Hemond, 1983, Rousk et al., 2013, 2015; Larmola et al., 2014; Vile et al., 2014; Diakova et al., 2016;
Stuart et al., 2021; Yin et al., 2022). Globally, peatlands store between 20 and 30 % of total soil carbon and
approximately 15 % of total soil nitrogen (Wieder and Vitt, 2006; Gallego-Sala et al., 2018; Fritz et al., 2014).
Microbial $N_2$-fixation helps to sustain C accumulation in peatlands and to remove carbon dioxide ($CO_2$) from the
atmosphere (Vile et al., 2014, and references therein). Changes in BNF may affect the dynamics of climate
change. A combination of high anthropogenic $N_r$ inputs with sustained $N_2$- fixation may accelerate invasion of
vascular plants into peat bogs leading to the reduction of the C–N stocks.
The nitrogen budget at the peat bog scale results from a balance between N inputs [atmospheric deposition of $N_r$,
mostly nitrate ($NO_3^-$) and ammonium ($NH_4^+$), with a contribution of organic N and BNF) and N outputs [runoff
dominated by dissolved, colloidal, and particulate N, and emissions of gaseous N forms, mainly nitrous oxide
($N_2O$), nitric oxide (NO), and $N_2$ as products of denitrification; Sgouridis et al., 2021]. The atmospheric lifetime
of $N_2O$, a potent greenhouse gas, is relatively long (>100 yr; Frolking et al., 2011). In contrast, the atmospheric
lifetime of NO, another greenhouse gas, is short (days), and, along with $N_2$ as the final product of denitrification
with no warming potential, is not considered in climate warming scenarios. Atmospheric deposition of $N_r$ in
high-latitude pristine bogs is 0.5-1.0 kg ha$^{-1}$yr$^{-1}$ (Vitt et al., 2003). Bogs receiving less than 10 kg $N_r$ ha$^{-1}$yr$^{-1}$are
defined as low-polluted (Lamers et al., 2000). Bogs receiving more than 18 kg $N_r$ ha$^{-1}$yr$^{-1}$are considered to be
highly polluted. Reactive N deposited on the surface of ombrotrophic peat bogs is vertically mobile (Novak et
al., 2014).
Nitrogen capture in rain-fed bogs is dominated by *Sphagnum* mosses (Limpens et al., 2006). Nitrogen-fixing
microbes (diazotrophs) mostly reside inside specialized *Sphagnum* cells (hyalocytes), although the mosses'

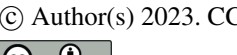


metabolic demands for N are supported also by free-living diazotrophs. In contrast, diazotrophs in feather
mosses, common in boreal forests, live epiphytically on the leaves (DeLuca et al., 2002; Rousk et al., 2015).
Endophytic diazotrophs are more protected against environmental fluctuations, including changes in $N_r$
deposition. BNF in bogs is associated mostly with cyanobacteria and methanotrophs (Larmola et al., 2014; Vile
et al., 2014; Leppanen et al., 2015; Holland-Moritz et al., 2021; Kolton et al., 2022). It follows that BNF may
affect potential methane ($CH_4$) emissions in two opposing directions: while higher C accumulation due to
efficient BNF may lead to higher $CH_4$ emissions during peat decomposition, $N_2$-fixing methanotrophs may
reduce emissions of $CH_4$ by oxidizing this greenhouse gas.

Recent work in peatlands has quantified the relative roles of various biotic and abiotic controls over BNF.
Leppanen et al. (2015) reported than BNF rates were independent of the diazotroph community structure. The
effect of temperature was reviewed by Carrell et al. (2019), Zivkovic et al., (2022), and Yin et al. (2022). The
optimal temperature for BNF is 20-30 °C (Zielke et al., 2005). Dry conditions are generally unfavorable for
BNF, but the moisture–BNF correlation tends to be insignificant (Yin et al., 2022). The effect of phosphorus (P)
as a limiting nutrient was evaluated by Limpens et al. (2004), Larmola et al. (2014), Ho and Bodelier (2015), van
den Elzen et al. (2017, 2020), and Zivkovic et al. (2022). In an interplay with other environmental and chemical
parameters, higher P availability may augment BNF. The role of the $NH_4^+/NO_3^-$ ratio in atmospheric deposition
as a BNF control was evaluated by Saiz et al. (2021). A higher $NH_4^+$ proportion relative to the total $N_r$ deposition
may result in lower BNF rates. Stuart et al. (2021) stressed a strong interaction between moss identity,
temperature, moisture and pH as possible BNF drivers. Kox et al. (2018) reported higher BNF rates under
oxygen ($O_2$) depletion. Wieder et al. (2019, 2020) and Kox et al. (2020) showed that BNF rates generally
increase in the presence of light.

The rates of BNF are measured using an acetylene reduction assay (ARA), $^{15}N_2$ isotope-labelling incubations, or
compound-specific amino acid $^{15}N$ probing (*e.g.,* Knorr et al., 2015; Chiewattanakul et al., 2022). Recent studies
have stressed the need for caution in ARA studies (Vile et al., 2014; Saiz et al., 2019; Soper et al., 2021).
Inhibition of the activity of methanotrophs by acetylene may lead to an underestimation of BNF rates. These
methods of direct measurements inevitably choose specific experimental conditions and thus provide *potential*
*instantaneous* BNF rates. A complementary, indirect evaluation of BNF can be  based on natural-abundance
$^{15}N/^{14}N$ isotope systematics (Novak et al., 2016; Zivkovic et al., 2017; Saiz et al., 2021; Stuart et al., 2021).
*Sphagnum* taking up N through BNF would carry a $\delta^{15}N$ signature close to 0 ‰, a value characterizing
atmospheric $N_2$ ($\delta^{15}N$ values are defined as a per mil deviation of the $^{15}N/^{14}N$ ratio in the sample from a standard;
the widely used standard is atmospheric $N_2$). With increasing BNF rates, the $\delta^{15}N$ values of living *Sphagnum*
converge from the often negative $\delta^{15}N$ value of atmospheric deposition to the 0 ‰ value of the source $N_2$. This
simple approach is complicated by tight inner N cycling near the bog surface, involving open-system isotope
fractionations. In particular, *Sphagnum* may additionally take up $N_r$ resulting from mineralization of organic N.
Because denitrification preferentially removes isotopically light N in a gaseous form, the residual $N_r$ in bog
water may become isotopically heavy and supply high-$\delta^{15}N$ nitrogen for assimilation. Mineralized $N_r$ in bog
water as another nutrient source may thus be isotopically similar to atmospheric $N_2$ (Novak et al., 2019; Stuart et
al., 2021).




BNF is an energy-intensive process requiring 16 adenosine-triphosphate (ATP) molecules to fix 1 mol of $N_2$. It
follows that, with an increasing input of pollutant $N_r$ *via* atmospheric deposition, BNF should be rapidly
downregulated. However, experiments applying additional $N_r$ to *Sphagnum* both in the laboratory and in the
field have indicated contradictory impacts on BNF. Some studies have shown a decrease in BNF rates in the
proximity of anthropogenic $N_r$ sources (Wieder et al., 2019; Saiz et al., 2021), while others have indicated
continuing BNF even at N-polluted sites (van den Elzen et al., 2018). BNF data from natural settings with
known time-series of historical $N_r$ deposition rates are rare (van den Elzen et al. 2018; Saiz et al., 2021). The aim
of the current study was to quantify BNF at high-elevation *Sphagnum*-dominated peatlands in an industrial part
of Central Europe, also known for intense agriculture. We combined $^{15}N_2$-tracer experiments with a natural
abundance N-isotope study at three peat bogs situated in the northern Czech Republic to provide qualitative
insights into the roles of individual BNF drivers. Our specific objectives were: (i) to investigate whether BNF
rates at the study sites correlate with well-constrained $NO_3^-$ and $NH_4^+$ deposition rates and P availability, and (ii)
to compare the results of experiments investigating $^{15}N$-assimilation by *Sphagnum* with the results of a natural-
abundance $\delta^{15}N$ inventory of individual wetland pools and fluxes. We expected that convergence of *Sphagnum*
N toward $\delta^{15}N_{N2} = 0$ ‰ would corroborate the relative magnitude of instantaneous BNF rates in between-site
comparisons.

**2. Materials and methods**

*2.1. Study sites*

The three studied *Sphagnum*-dominated peat bogs (Fig. 1, Tab. 1) are located in the northern Czech Republic, a
highly industrialized part of Central Europe with numerous coal-burning power plants. In the 1970s-1990s,
Norway spruce monocultures were affected by acid rain in the vicinity of Brumiste (BRU; Krusne Mts.) and
Uhlirska (UHL; Jizerske Mts.). At UHL, most spruce stands died back and were harvested. The third site, Male
Mechove Jezirko (MMJ; Jeseniky Mts.) is surrounded by relatively healthy mature spruce forests. The distance
between adjacent study sites is 160-190 km (Fig. 1). The studied high-elevation catchments are drained by small
streams. The studied peatlands are partly rain-fed, with a possible contribution of lateral water influx from the
surrounding segments of the catchments. The bedrock is composed of granite at BRU and UHL, and phyllite at
MMJ. The surface of each bog is characterized by a combination of hummock–hollows microtopography and
lawns (Dohnal, 1965). Moss species at BRU include *S. cuspidatum,* common in hollows and pools, *S.*
*magellanicum,* mostly occupying intermediate positions between the tops of the hummocks and the hollows, *S.*
*rubellum,* typical of dense carpets in rain-fed bogs, and *S. papillosum,* forming low hummocks and mats in bogs
and mires. At UHL and MMJ, the predominant moss species is shade-demanding *S. girgensohnii*, requiring
slight base enrichment (Tab. S1 in the Supplement). The growing season is more than seven months long, from
late March to early November. The measured density of living *Sphagnum* is 0.04 g cm$^{-3}$. More details on BRU
are in Bohdalkova et al. (2013), and Buzek et al. (2019, 2020). Biogeochemical processes at UHL were studied
by Novak et al. (2005), Sanda and Cislerova (2009), Bohdalkova et al. (2014), Marx et al. (2017), Oulehle et al.
(2017, 2021a), and Vitvar et al. (2022). Further information on MMJ is in Novak et al. (2003, 2009).



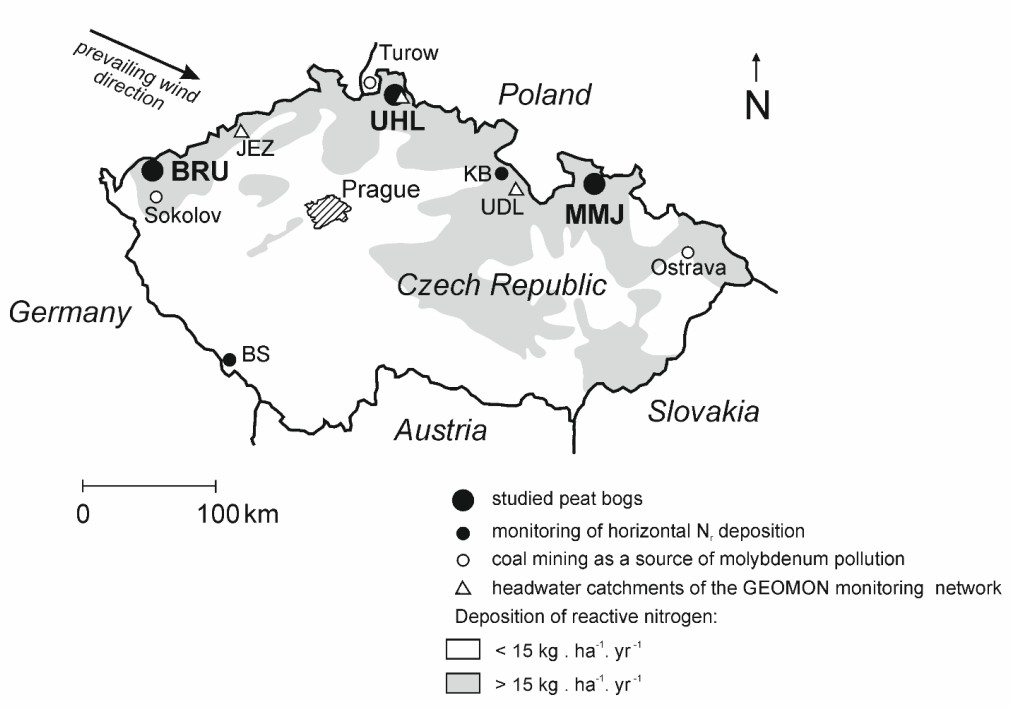

**Fig. 1.** Location of the studied *Sphagnum*-dominated peat bogs.

**Table 1.** Study site characteristics.

| Site | Location | Elevation (m) | Long-term precipitation total (mm yr⁻¹) | Mean annual temperature (ºC) | Bog area (ha) | Maximum peat depth (cm) | Atmospheric vertical $N_r$ deposition (kg ha⁻¹ yr⁻¹)[1] | Total atmospheric $N_r$ deposition (kg ha⁻¹ yr⁻¹)[2] | $NH_4^+$-N/$NO_3^-$-N ratio |
|---|---|---|---|---|---|---|---|---|---|
| Brumiste, BRU | 50º24′ N 12º36′ E | 930 | 1080 | 4.5 | 17 | 200 | 12.7 | 16.5 | 1.2 |
| Uhlirska, UHL | 50º49′ N 15º08′ E | 830 | 1230 | 4.0 | 50 | < 200 | 15.5 | 20.2 | 1.2 |
| Male Mechove jezirko, MMJ | 50º13′ N 17º18′ E | 750 | 1090 | 5.3 | 195 | 660 | 14.3 | 18.6 | 1.3 |

[1]long-term average according to Oulehle et al., 2016
[2]including 30 % of horizontally deposited $N_r$ (Novak et al., 2015)

*2.2. Sampling*



Samples of rain and snow for $\delta^{15}N$ determinations were collected between January 2016 and October 2019 using
a simplified protocol of Fottova and Skorepova (1998). Open-area precipitation was sampled by two rain
collectors placed five meters apart, 160 cm above ground. Spruce canopy throughfall was sampled using five
(UHL) or three (BRU, MMJ) collectors installed 10 m apart. Deposition samplers were polyethylene (PE)
funnels (surface area of 113 $cm^2$) fitted to 1-L bottles. In winter, cylindrical PE vessels (surface area of 167 $cm^2$)
were used to collect snow. At the end of cumulative one-month sampling, open area precipitation and throughfall
samples, respectively, were pooled prior to chemical and N-isotope analysis. One-liter samples of runoff were
collected in ~30-day intervals at BRU over a 25-month period, samples of runoff were collected at UHL and
MMJ in summer 2019 (*see* Tab. S2 for specific dates). Five replicates of surface bog water were collected
throughout each study site in June 2019. The depth of the water pools was less than 20 cm. The total number of
water samples for $\delta^{15}N$ analysis was 136.
A vertical peat core, 10-cm in diameter, 30-cm deep, was collected in a *Sphagnum*-dominated lawn at each of
the study sites in October 2018, kept vertically at 6 °C for 12 hours and then frozen. At the same time, 12
samples of living *Sphagnum* were collected randomly throughout each bog for species identification and N
isotope analysis. Additionally, 12 replicate samples of living *Sphagnum* were collected in various parts of each
of the peat bogs for a $^{15}N_2$-labelling experiment. Each replicate sample consisted of 30 individual 5-cm long
*Sphagnum* plants. *S. girgensohnii* was used in the UHL and MMJ experiments, a mix of *S. magellanicum, S.*
*papillosum,* and *S. cuspidatum* was used in the BRU experiment (*cf.,* Tab. S1); *Sphagnum* samples were
transported to the laboratory at a temperature of 6 °C.
*2.3. $^{15}N_2$ Sphagnum incubation experiment*
Measurements of potential $N_2$-fixation rates were performed using a modified protocol of Larmola et al. (2014).
Four plant replicates per site were analyzed at time t = 0 without incubation (control no. 1). Eight replicates per
site were closed in 200-mL transparent PE containers with 5 mL of bog water collected at BRU, UHL and MMJ,
respectively. Twenty-four mL of headspace air were removed from four replicates in closed containers and
replaced with 32 mL of $^{15}N_2$ tracer gas containing 98 atomic % of $^{15}N$ (Aldrich, Germany). The four remaining
*Sphagnum* replicates with no $^{15}N_2$ addition served as a procedural control no. 2 to identify possible incubation
artifacts. The $^{15}N$-labelled and control-no.-2 replicates were incubated for 168 hours. Each day, the temperature
in the growth chamber was kept at 18 °C for 16 hours at daylight, and at 10 °C for 8 hours under dark conditions.
Following N-isotope analysis, BNF rates were calculated according to Vile et al. (2014) and Knorr et al. (2015):
$$N_{2fix} = \frac{\Delta\,at.\,\%\,^{15}N_{Sph}}{\Delta\,at.\,\%\,^{15}N_{gas}} \; x \; \frac{total\ N\ \%_{Sph}}{t * 100} \quad (g\ N\ g\ DW^{-1}\ day^{-1}), \hspace{2cm} (1)$$
where $N_{2fix}$ is the $N_2$-fixation rate in g N g $DW^{-1}$(*Sph)* $day^{-1}$, t is incubation time (days), total N% $_{Sph}$, $\Delta$ at. %
$^{15}N_{Sphagnum}$ is the difference between atom % labeled and control sample, $\Delta$ at. % $^{15}N_{gas}$ is the difference between
the concentration $^{15}N$ in the headspace and the natural abundance (at. %). The used *Sphagnum* density was 0.04
g $cm^{-3}$.






We note that our $^{15}N_2$ experimental design used a longer incubation period (168 hr), compared to most previous
studies (24 to 80 hr; *cf.,* Knorr et al., 2015). To minimize the effect of changing headspace concentrations of $O_2$
and $CO_2$ on the living moss and the microbiome, we used larger sealed containers, compared to most previous
studies (200 *vs.* ≤ 125 mL). It also bears mention that Dabundo et al. (2014) found a deviation from the declared
$^{15}N_2$ purity within commercially available tracer tanks. We did not study the tracer purity and hence the observed
BNF rates might be viewed as maximum estimates. Because our incubation study was based on one-time
measurements under laboratory conditions, in the current paper we chose not to upscale the BNF rates to the
entire peat bog and an annual time span.

*2.4. Chemical and isotope analysis*

Frozen peat cores were sectioned to 2-cm thick segments. Samples of peat and *Sphagnum* were dried and
homogenized. Nitrogen concentrations in peat and *Sphagnum* samples were determined on a Fisons 1180
elemental analyzer with a 1.5 % reproducibility (2σ). Ammonium and nitrate concentrations in water samples
were determined spectrophotometrically with a reproducibility of 0.1 mg L$^{-1}$. About 0.5 L of each water sample
were used to separate $NH_4^+$ and $NO_3^-$ (Bremner, 1965). Nitrogen isotope composition was measured on a Delta
V mass spectrometer and expressed in $\delta^{15}N$ notation. IAEA isotope standards N1 ($\delta^{15}N$ = 0.4 ‰) and N2 ($\delta^{15}N$ =
20.3 ‰) were analyzed before every session, and two in-house standards (ammonium sulfate, $\delta^{15}N$ = -1.7 ‰,
and glycine, $\delta^{15}N$ = 4.0 ‰) were analyzed after every six samples. The reproducibility of the $\delta^{15}N$
determinations was 0.30 and 0.35 ‰, for the liquid and solid samples, respectively. Methods of concentration
analysis of other chemical species are given in *Appendix I.*

*2.5. Historical rates of $N_r$ deposition*

Long-term data from 32 monitoring stations in the Czech Republic operated by the Czech Hydrometeorological
Institute, Prague, were used to assess temporal and spatial variability of $NH_4^+$ and $NO_3^-$ concentrations in vertical
deposition using a model by Oulehle et al. (2016). Median *z*-score values of $NH_4^+$ and $NO_3^-$ concentrations
derived from observations at the monitoring stations and nation-wide emission rates, published by Kopacek and
Vesely (2005), and Kopacek and Posh (2011), showed significant relationships at the *p* < 0.001 level. Using
linear models, *z*-score values were expressed for the period 1900-2012 and then back-transformed to give
concentration estimates for the study sites. Annual rates of vertically deposited $NH_4^+$ and $NO_3^-$ were products of
modelled concentrations and precipitation quantities at BRU, ULH and MMJ.

*2.6. Statistical evaluation*

Statistical analysis was performed using the R software (R Core Team, 2019) version 3.6.2, and its contributed
packages *sandwich* (Zeileis, 2004) and *multcomp* (Hothorn et al., 2008). Comparisons of groups of N isotope
and concentration data (see sections 2.3 and 2.4)



were based on one-way analysis of variance with a sandwich estimator of covariance matrix to account for
heteroscedasticity among the groups (MacKinnon and White, 1985). *Post-hoc* multiple comparisons of the same
groups were then performed according to Hothorn et al. (2008). Because of the largely uneven number of runoff
samples *per* site (50, 6, and 2 at BRU, UHL and MMJ, respectively), we did not include runoff $\delta^{15}$N data in the
statistical evaluation.

**3. Results**

*3.1. Historical rates of atmospheric $N_r$ inputs*

Vertical deposition of $NH_4^+$ reached a maximum in 1980, remained almost unchanged until 1990, and decreased
thereafter (Fig. 2a). Nitrate-N deposition exhibited a wider maximum between *ca.* 1970 and 1990 (Fig. 2b). In
the oldest modelled time period (1900-1930), ammonium in deposition dominated over nitrate. During the
deposition peak, the contributions of $NH_4^+$-N and $NO_3^-$-N to total vertical $N_r$ deposition were similar (8 to 13 kg
ha$^{-1}$ yr$^{-1}$ at individual sites). Across the modelled years, the $NH_4^+$-N/$NO_3^-$-N ratio in vertical deposition was
similar at all three sites (1.2 to 1.3; Tab. 1). Since *ca.* 1950, pollution at the study sites *via* total vertical
deposition of inorganic $N_r$ increased in the order: BRU < MMJ < UHL (Fig. 2c). Fig. 2c shows that the between-
site differences in the most recent years have been small (1-2 kg N ha$^{-1}$ yr$^{-1}$).



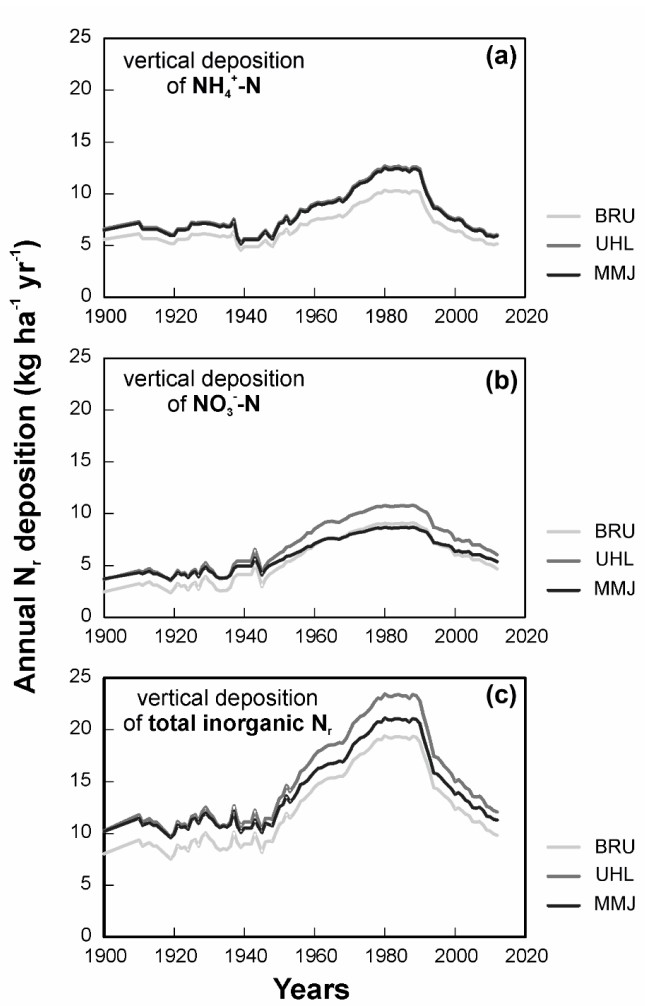


**Fig. 2.** Modelled long-term changes in atmospheric $N_r$ deposition according to Oulehle et al. (2016).


*3.2. $^{15}N_2$ incubation experiment*

There were no statistically significant differences between $\delta^{15}N$ values of *Sphagnum* at time $t = 0$ and at time $t =$
168 hours following incubation in natural atmosphere (controls no. 1 and 2; Tab. 2; $p > 0.05$). Mean $\delta^{15}N$ values
of the moss of the two controls were similar among the sites (-3.6 to -4.1 ‰). At the end of the $^{15}N_2$ *Sphagnum*
incubation, there was no change in the N isotope signature of the moss at BRU and UHL ($p > 0.05$). In contrast,
there was a large positive shift in $\delta^{15}N$ values of *Sphagnum* collected at MMJ (59.2 to 467 ‰; Tab. 2; Fig. 3).
The $N_2$ fixation rate calculated from the N isotope systematics in the $^{15}N_2$ labelling experiment was 0 at BRU
and UHL, and 4.11 μg N g$^{-1}$ d$^{-1}$, or 8.20 mg N m$^{-2}$ d$^{-1}$ at MMJ.



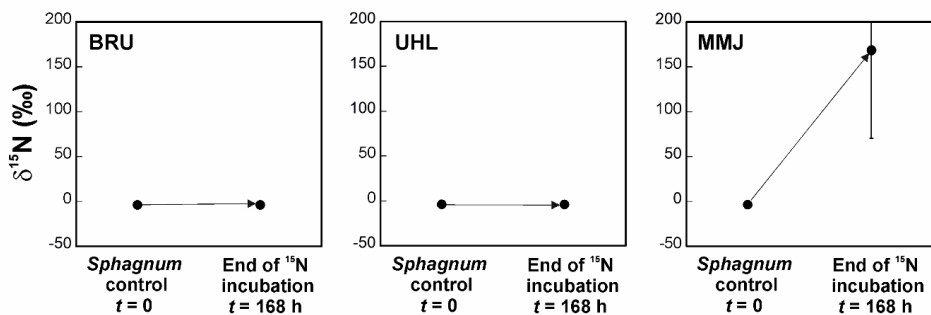


**Fig. 3**. Results of a $^{15}N_2$ incubation study using living *Sphagnum*. Means and standard errors are given.

**Table 2**. Positive $\delta^{15}N$ shift in total moss nitrogen following the $^{15}N_2$ assay incubation at MMJ.

| Site | BRU | | | UHL | | | MMJ | | |
|---|---|---|---|---|---|---|---|---|---|
| | $\delta^{15}N$ (‰) | | | | | | | | |
| | *Sphagnum* control $t_0$ | *Sphagnum* control $t = 168$ h | *Sphagnum* at the end of $^{15}N_2$ incubation $t = 168$ h | *Sphagnum* control $t_0$ | *Sphagnum* control $t = 168$ h | *Sphagnum* at the end of $^{15}N_2$ incubation $t = 168$ h | *Sphagnum* control $t_0$ | *Sphagnum* control $t = 168$ h | *Sphagnum* at the end of $^{15}N_2$ incubation $t = 168$ h |
| Replicate 1 | -3.9 | -4.0 | -4.1 | -3.7 | -3.8 | -3.9 | -2.7 | -2.7 | 467 |
| Replicate 2 | -3.9 | -4.1 | -3.9 | -3.9 | -3.7 | -3.7 | -4.0 | -3.8 | 59.2 |
| Replicate 3 | -3.9 | -4.2 | -4.3 | -4.4 | -4.0 | -4.2 | -3.8 | -4.0 | 68.8 |
| Replicate 4 | -3.5 | -3.8 | -3.6 | -4.7 | -4.6 | -4.6 | -3.8 | -4.2 | 83.0 |
| Mean ± SE | -3.8 ± 0.1 | -4.0 ± 0.1 | -4.0 ± 0.2 | -4.1 ± 0.2 | -4.0 ± 0.2 | -4.1 ± 0.2 | -3.6 ± 0.3 | -3.7 ± 0.4 | 169 ± 99.2 |


*3.3. Natural-abundance N-isotope systematics*

*3.3.1. Atmospheric deposition*

Ninety-six per cent of the deposited inorganic $N_r$ species had negative $\delta^{15}N$ values; *i.e.,* contained isotopically
light N (Tab. S2; Fig. S1). The mean $\delta^{15}N$ value across all 181 samples of atmospheric deposition was $-5.3 \pm 0.3$
‰ (SE). Mean $\delta^{15}N$ values of both forms of atmospherically deposited N ($NH_4^+$ and $NO_3^-$) in an open area were
slightly higher than those in throughfall at BRU and MMJ, and slightly lower than those in throughfall at UHL
(Tab. 3). Nitrate-N in open-area deposition was on average slightly isotopically lighter than $NH_4^+$-N at all three
sites. At the 0.05 probability level, however, the within-site differences among deposition sample types and
among N species at BRU and MMJ were insignificant. The only statistically significant difference was found
between $\delta^{15}N$ values of open-area $NO_3^-$ and both N species in throughfall at UHL (*see* superscript letters in Tab.

3).




**Table 3**. Multiple comparisons among $\delta^{15}$N values of four sample types of atmospheric deposition. Different
letters in superscript denote statistical difference ($p < 0.05$).

| | mean $\delta^{15}$N (‰) ± SD | | |
| --- | --- | --- | --- |
| Site | BRU | UHL | MMJ |
| open-area $NH_4^+$ | $-5.18 \pm 3.63^a$ | $-5.84 \pm 3.31^{ab}$ | $-3.48 \pm 6.01^a$ |
| open-area $NO_3^-$ | $-5.71 \pm 2.82^a$ | $-6.19 \pm 2.34^b$ | $-4.10 \pm 3.18^a$ |
| throughfall $NH_4^+$ | $-6.86 \pm 3.10^a$ | $-3.15 \pm 1.66^a$ | $-6.57 \pm 6.40^a$ |
| throughfall $NO_3^-$ | $-6.16 \pm 2.29^a$ | $-4.17 \pm 0.58^a$ | $-6.02 \pm 4.14^a$ |



*3.3.2. Comparison of $\delta^{15}$N values of Sphagnum and atmospheric deposition*

The $\delta^{15}$N values of living *Sphagnum* were between -6.2 and -1.9 ‰ (Tab. S1). The $\delta^{15}$N values of living
*Sphagnum* at BRU were statistically different from the $\delta^{15}$N values of atmospheric deposition (means of -4.0 and
-5.9 ‰, respectively; $p < 0.05$; Fig. 4). At UHL (means of -4.3 and -5.6 ‰, respectively;) and MMJ (means of -
4.4 and -4.3 ‰, respectively), the differences between the $\delta^{15}$N values of living *Sphagnum* and the $\delta^{15}$N values
of atmospheric deposition were insignificant ($p > 0.05$; Fig. 4). At BRU (but also at UHL), *Sphagnum* N was on
average isotopically heavier than deposited N, *i.e.,* closer to the 0 ‰ value of atmospheric $N_2$. Nitrogen
concentration in living *Sphagnum* was significantly higher at MMJ (mean of 1.0 wt. %) than at UHL (0.9 wt. %;
$p < 0.05$; Fig. 5). The mean N concentration in BRU *Sphagnum* was 1.0 wt. %, indistinguishable from the other
two study sites.



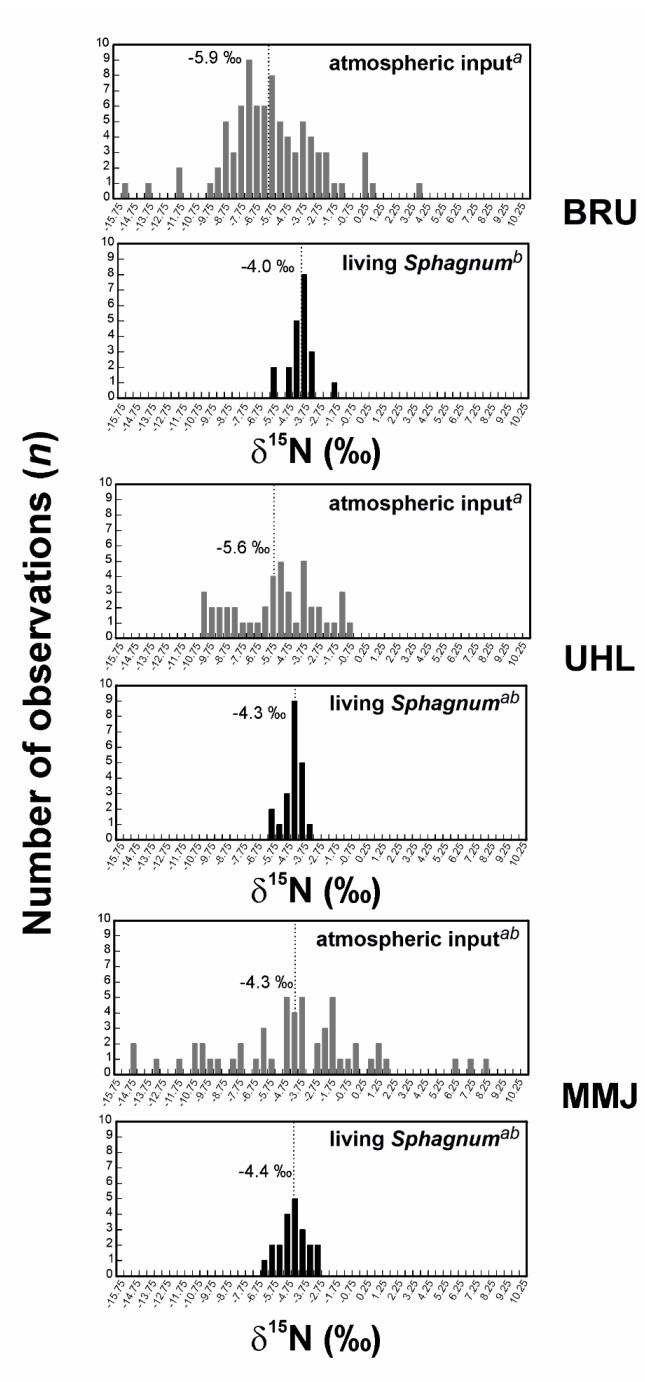

**Fig. 4.** Histograms of $\delta^{15}N$ values of atmospheric input of $N_r$ and living *Sphagnum*. Different letters in superscript mark statistically different sample types ($p < 0.05$).



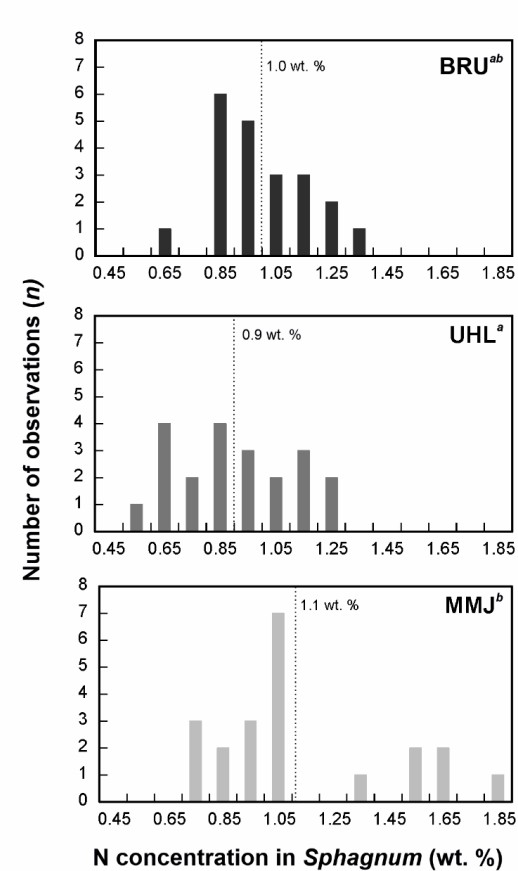

**Fig. 5.** Nitrogen concentrations in living *Sphagnum*. Different letters in superscript mark statistically different
sample types ($p < 0.05$).

*3.3.3. Multiple $\delta^{15}N$ comparisons among sample types*

The mean $\delta^{15}N$ value of surface bog water was 0.9 ‰ at BRU, 1.8 ‰ at UHL, and -1.9 ‰ at MMJ. Nitrogen in
surface bog water was isotopically significantly heavier than N in both *Sphagnum* and atmospheric input at all
three sites (Fig. 6; $p < 0.05$). At BRU and UHL, the mean $\delta^{15}N$ value of surface bog water was higher than the 0
‰ value of atmospheric $N_2$. At MMJ, the mean $\delta^{15}N$ value of surface bog water was lower than the N isotope
signature of atmospheric $N_2$. In other words, all three sample types (deposition, *Sphagnum*, and bog water) at
MMJ contained isotopically lighter N, compared to atmospheric $N_2$ (Fig. 6).

When averaged across all depths (0-30 cm), the mean $\delta^{15}N$ value in the peat core was -2.4 ‰ at BRU, -0.4 ‰ at
UHL, and -1.9 ‰ at MMJ. At all three sites, the maturing peat in the vertical profile contained isotopically
significantly heavier N compared to living *Sphagnum* ($p < 0.05$; Fig. 6; Tab. S2).






The mean $\delta^{15}N$ value of runoff was -2.7 ‰ at BRU (combined $NH_4^+$ and $NO_3^-$ data; number of observations $n =$
50), -5.3 ‰ at UHL ($n = 6$), and -5.1 ‰ at MMJ ($n = 2$; Tab. S1). The N isotope signature of runoff was higher
compared to the atmospheric input at BRU, and similar with the atmospheric input at UHL and MMJ (small
solid squares in Fig. 6). At all three sites, runoff contained isotopically lighter N compared to bog water (Fig. 6).

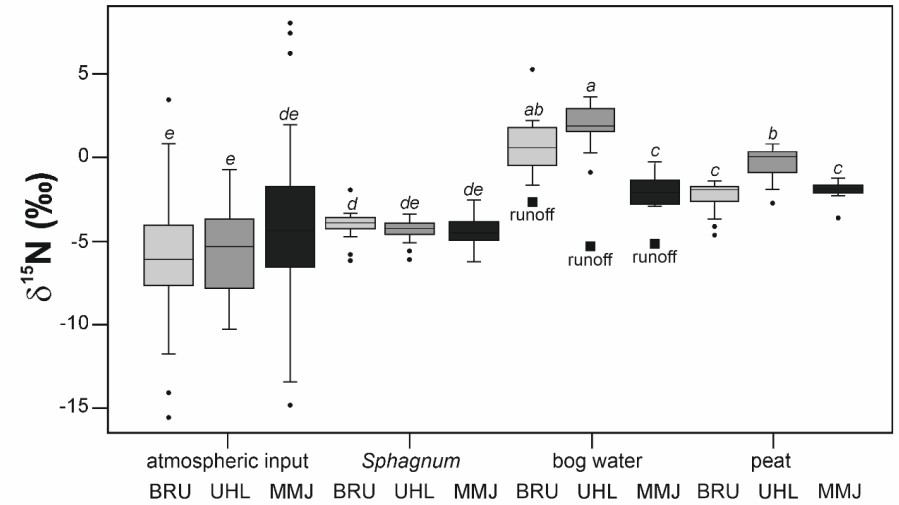


**Fig. 6.** Between-site comparisons of $\delta^{15}N$ values of studied N pools and fluxes. Horizontal lines in boxes
correspond to median values. Different letters mark statistically different sample types ($p < 0.05$).

*3.4. Chemistry of natural waters*

*Acidity.* Surface bog water had lower pH than atmospheric deposition and runoff at all three sites. Mean bog
water pH was 4.0 at UHL, 4.3 at BRU, and 4.9 at MMJ (Tab. S3). The pH of atmospheric deposition was lower
than 5.0 only at UHL.

*Nitrogen.* The maximum $NH_4^+$-N concentration in open area precipitation was 1.7 mg $L^{-1}$ (UHL) and the
maximum $NO_3^-$-N concentration in open area precipitation was 7.1 mg $L^{-1}$ (BRU; Tab. S2). The maximum
concentration of $NH_4^+$-N in throughfall was 3.9 mg $L^{-1}$(MMJ) and the maximum concentration of $NO_3^-$-N in
throughfall was 9.7 mg $L^{-1}$ (BRU; Tab. S2). The maximum concentration of $NH_4^+$-N in bog water was 2.3 mg $L^-$
$^1$(UHL) and the maximum concentration of $NO_3^-$-N in bog water was 2.7 mg $L^{-1}$ (MMJ; Tab. S2). The maximum
concentration of $NH_4^+$-N in runoff was 1.3 mg $L^{-1}$ (BRU) and the maximum concentration of $NO_3^-$-N in runoff
was 7.1 mg $L^{-1}$ (BRU; Tab. S2).

*Phosphorus.* The mean concentration of total P in atmospheric deposition increased in the order: BRU (below
6.0 µg $L^{-1}$) < UHL (9.3 µg $L^{-1}$) < MMJ (15.5 µg $L^{-1}$; Tab. S3). Phosphorus concentration in surface bog water



was roughly 30 times higher than in atmospheric deposition at BRU, more than 50 times higher at UHL, and
more than 10 times higher at MMJ (Tab. S3). The UHL bog water contained as much as 490 μg P L$^{-1}$. The mean
P concentration in runoff increased in the order: MMJ (12.4 μg L$^{-1}$) < BRU (29.4 μg L$^{-1}$) < UHL (40.2 μg L$^{-1}$;
Tab. S3).

*Other chemical species.* Natural waters at UHL were richer in sulfate ($SO_4^{2-}$) than those at the remaining two
sites (Tab. S3). UHL bog water and runoff contained as much as 47.4 and 33.7 mg $SO_4^{2-}$ L$^{-1}$, respectively. Bog
water was richer in potassium ($K^+$) at UHL (9.05 mg L$^{-1}$) compared to BRU and MMJ (1.85 and 1.97 mg L$^{-1}$,
respectively). The concentration of DOC in atmospheric deposition was 2-4 times higher at MMJ than at the
remaining two sites (Tab. S3). In contrast, surface bog water at MMJ had 1.4 to twice lower DOC
concentrations, compared to the remaining two sites. Detailed water chemistry is in Tab. S3.

*3.5. Vertical peat profiles*

From peat surface to the depth of 15 cm, peat density exhibited a slight increase similar at the three sites (Fig.
7a). Deeper, peat density remained relatively low (~0.05 g cm$^{-3}$) at MMJ, and continued increasing irregularly at
BRU and UHL. Ash content remained below 5 wt. % to a depth of 30 cm at MMJ, and, with one exception, also
at BRU (Fig. 7b). The highest ash content was observed at UHL. Below the depth of 20 cm, it increased
downcore to values greater than 10 wt. %. The total N concentrations in peat substrate increased downcore or
exhibited a zigzag pattern (Fig. 7c). The UHL peat core was the richest in N in most 2-cm peat sections. Down to
a depth of 15 cm, N concentration was the lowest in MMJ peat. At all three sites, the vertical $\delta^{15}N$ profile was
characterized by a downcore increase near the surface flattening out in the deepest peat sections (Fig. 7d).
Generally, the $\delta^{15}N$ values in peat cores increased in the order BRU < MMJ < UHL.

The nearly constant carbon (C) concentrations in peat were similar at all three sites to the depth of 20 cm, and
became more variable deeper (Fig. 7e). The sharpest downcore decrease in the C:N ratio was found at MMJ,
with the exception of the 0-to-4 cm depth where the C:N ratio increased (7f). Throughout the vertical peat
profiles, P concentration was the lowest at BRU, and the highest at UHL (Fig. 7g). The N:P ratio was close to 12
throughout the UHL peat profile, increased downcore at MMJ from 10 to 20, and exhibited an irregular pattern
at BRU, ranging between 20 and 40 (Fig. 7h). Further information on vertical changes in peat composition is in
Tab. S4.



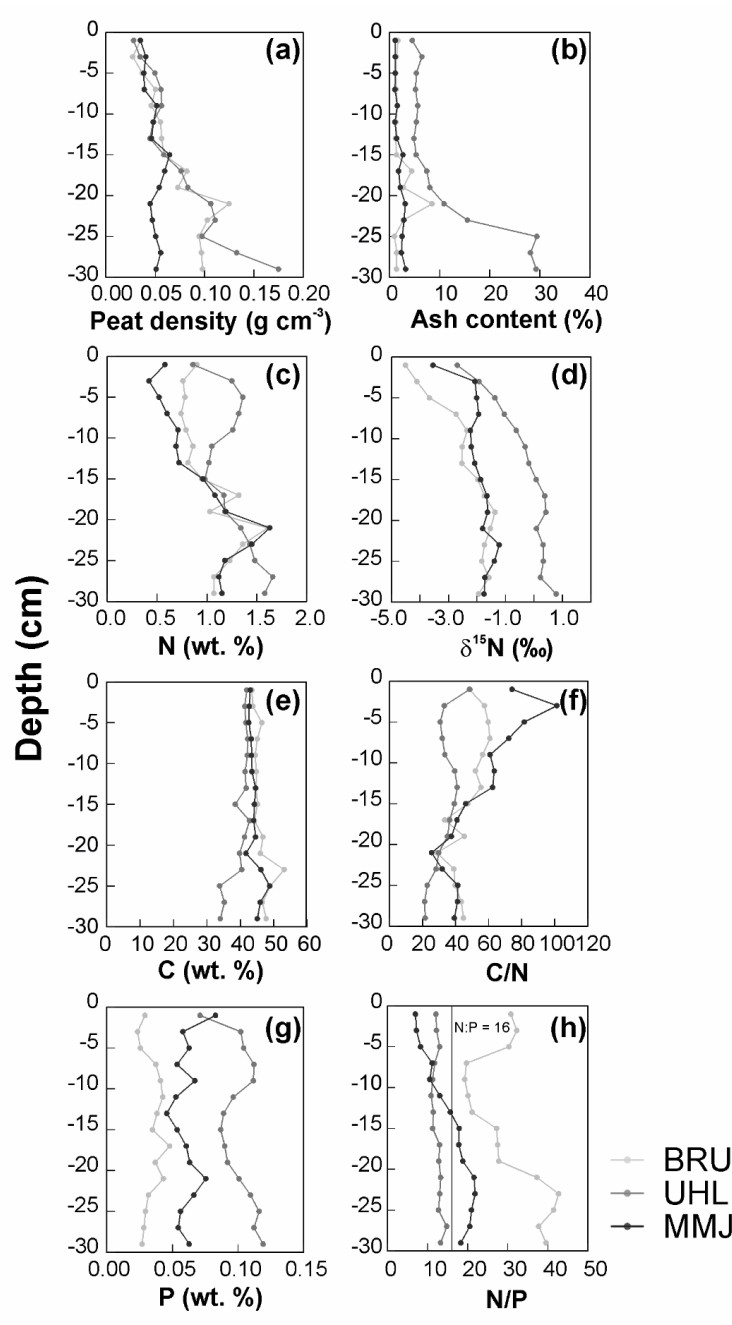


**Fig. 7.** Vertical changes in physicochemical characteristics of *Sphagnum* peat.




**4. Discussion**

*4.1. The role of horizontal $N_r$ deposition in peatlands*

Using field experiments, we have recently shown a sizeable contribution of horizontally deposited $N_r$ to total atmospheric deposition in Central European *Sphagnum* peat bogs (Novak et al., 2015b). During 80-90 days of the spring and fall foggy seasons, horizontal deposition added another 45 % to vertical deposition at Kunstatska Kaple Bog (KB), a mountain-top site in northern Czech Republic, and 14 % at Blatenska Slat (BS) in the less polluted southern Czech Republic (*see* Fig. 1 for location). Additionally, Hunova et al. (2023) reported a relatively high horizontal contribution of nitrate-N to winter-time atmospheric deposition in Czech mountains by analyzing ice accretions (mean of $29 \pm 3$ %; data for December–March; number of sites $n = 10$). As a first approximation, we suggest that the upper limit of the contribution of horizontal deposition to vertical deposition at BRU, UHL and MMJ could have been 30 %. If so, the total average $N_r$ deposition was slightly higher than 18 kg ha$^{-1}$ yr$^{-1}$ at UHL and MMJ, and 16.5 kg ha$^{-1}$ yr$^{-1}$ at BRU (Tab. 1). Our study sites can thus be considered as highly or medium-polluted (Lamers et al., 2000). The overall $N_r$ pollution decreased in the order UHL > MMJ > BRU.

We note that total atmospheric deposition may also contain measurable amounts of total organic N (TON; Violaki et al., 2010; Cornell, 2011). TON fluxes have not been considered as part of the $N_r$ input in existing peatland BNF studies. Open-area precipitation at BRU, UHL and MMJ contained an additional 15, 45, and 13 % of total organic N, respectively, relative to the sum of the two inorganic $N_r$ forms (Tab. S3; October 2018). More TON data in precipitation would be needed to realistically estimate annual deposition of organic N at our study sites.

*4.2. Relationship between $N_r$ pollution and $N_2$-fixation*

In theory, chronic atmospheric deposition of pollutant $N_r$ should suppress BNF in peatlands (Wieder et al., 2019, 2020). Saiz et al. (2021) quantified downregulation of BNF along a geographical pollution gradient. Relative to a practically unpolluted site receiving 2 kg $N_r$ ha$^{-1}$ yr$^{-1}$ from the atmosphere, these authors reported a 54 % decrease in BNF rates under the atmospheric deposition of 6 kg $N_r$ ha$^{-1}$ yr$^{-1}$, a 69 % decrease under the deposition of 17 kg $N_r$ ha$^{-1}$ yr$^{-1}$, and a 74 % decrease under the deposition of 27 kg $N_r$ ha$^{-1}$ yr$^{-1}$. As seen in Fig. 3, our data did not confirm such an inverse correlation at Central European sites. Instead, the most and least polluted peat bog exhibited no instantaneous BNF, while MMJ, whose $N_r$ inputs were lower than those at UHL and higher than those at BRU, showed a high mean BNF rate. Given that most previous studies of *Sphagnum* bogs reported non-zero BNF rates regardless of atmospheric $N_r$ deposition level (*see* compilation in Tab. S5), non-detectable BNF rates at BRU and UHL were surprising. The mean instantaneous BNF rate at MMJ was lower than BNF rates in unpolluted high-latitude bogs in Canada (Vile et al., 2014) and Patagonia (Knorr et al., 2015). Among the studies listed in Tab. S5, the mean BNF rates at MMJ were the fourth highest. Our data from MMJ are consistent with a conclusion by Saiz et al. (2021) who suggested a development of diazotrophic microbes' tolerance to high rates of atmospheric $N_r$ deposition in recent decades. Global assessments of the



dependence of BNF on total $N_r$ deposition are difficult to make for several reasons: (i) few studies consider
horizontal $N_r$ deposition which may be sizeable and depends not just on atmospheric pollution, but also on
elevation; few studies have quantified atmospheric input of organic N (ii) there is a large within-site
heterogeneity in BNF ($^{15}N_2$ incubations should be performed using a large number of replicates, *see* $\delta^{15}N$
differences between individual MMJ replicates in Tab. 2; *cf.,* "BNF hotspots" in Stuart et al., 2021); and (iii)
recalculation between two commonly used BNF units (μg N per 1 g of *Sphagnum* $d^{-1}$, g N $m^{-2}$ $yr^{-1}$) in literature
data requires information on additional site-specific parameters, such as peat density, seasonality in daily
temperatures and snow cover duration. Additionally, it is often unclear to what maximum depth in peat bogs
BNF proceeds and whether there is a gradient in BNF rates within this depth range (Vile et al., 2014; Knorr et
al., 2015).

Since the differences in $N_r$ deposition among sites were minor (Tab. 1; Fig. 2), we suggest that $N_r$ deposition was
not the primary control of the BNF rates in our study at the time of *Sphagnum* sampling.

*4.3. Chemical and environmental parameters as possible BNF controls*

*4.3.1. The role of the $NH_4^+$-N/$NO_3^-$-N ratio in atmospheric deposition*

The impact of the two main $N_r$ forms in deposition on BNF can be different. Because BNF generates $NH_4^+$, the
need for BNF to complement metabolic demands of the moss may be lower if deposition of $NH_4^+$-N exceeds the
deposition of $NO_3^-$-N (van den Elzen et al., 2018; Saiz et al., 2021). At our study sites, the $NH_4^+$-N/$NO_3^-$-N ratios
were nearly identical (Tab. 1), slightly exceeding 1. It follows that this ratio was unlikely the driver of higher
BNF potential at MMJ, compared to the remaining two sites.

*4.3.2. The effect of temperature*

MMJ is situated at a lower elevation, compared to UHL and BRU, and its mean annual temperature is higher
than at the remaining two sites (Tab. 1). This could positively affect the rate of BNF (Basilier et al., 1978;
Schwintzer et al., 1983; Urban and Eisenreich, 1988; Zivkovic et al., 2022; Yin et al., 2022). By contrast, Carrell
et al. (2019) argued that BNF rates may decrease with an increasing temperature due to lower microbial diversity
and greater mineralization rates leading to more $N_r$ in bog water and hence lower demand for BNF. Under field
conditions of the Czech sites and at the peatland scale, temperature likely is a key factor regulating BNF. In our
$^{15}N$ assimilation study, however, the chosen temperature was identical for all three sites. Consequently,
temperature was not the dominant control of the measured short-term BNF rates.

*4.3.3. The effect of bog wetness*

Fig. S2 shows monthly measurements of water table level below bog surface at BRU (Bohdalkova et al., 2013)
and UHL (Tacheci, 2002). The mean annual water table depth was -5.2 ± 2.3 and -7.5 ± 1.1 cm at BRU at UHL,
respectively. No water level monitoring data are available for MMJ, however, during our field sampling



campaigns, numerous 10-to-20 cm deep water pools were observed near the bog center at MMJ, especially
during the growing seasons of 2017 and 2019. Other high-elevation peat bogs on crystalline bedrock previously
studied in the Czech Republic exhibited water table fluctuation at shallow depths of 5-8 cm, similar to BRU and
MMJ (Novak and Pacherova, 2008). Based on visual inspection, somewhat drier conditions were typical of
UHL, compared to the other two sites. Hydrological monitoring (GEOMON network database, Czech
Geological Survey; Oulehle et al., 2021b) revealed significantly drier conditions at UHL in the water year 2018,
compared to the long-term average given in Tab. 1. Precipitation totals at UHL were 1460 mm in 2016, 1370
mm in 2017, mere 892 mm in 2018, and 1230 mm in 2019. The ecosystem suffered from chronic drought in
2018 also at other GEOMON sites, JEZ (the nearest site to BRU) and UDL (the nearest site to MMJ; for location
*see* Fig. 1). While *Sphagnum* for the $^{15}N_2$ incubation was collected at all three study sites at the same time
(October 2018), site-specific moisture conditions could have affected microbial community structure and the
BNF potential. In the laboratory experiment, however, similar wetness was ensured by the same volume of
added bog water to *Sphagnum* from all three sites. Therefore, we suggest that water availability did not control
the instantaneous BNF rates.

*4.3.4. The effect of Sphagnum species*

Stuart et al. (2021) showed that host identity is often the primary driver of BNF in peatlands. Under low $N_r$
pollution, higher species-specific litter decomposability augments BNF by increasing nutrient turnover (van den
Elzen et al., 2020). Saiz et al. (2021) observed higher BNF rates in *Sphagnum* species typical of hollows than
those dominating hummocks. Specifically, *S. fallax* exhibited higher BNF rates than *S. capillifolium and S.*
*papillosum.* The reason for such systematics appeared to be that the anoxic environment of wet hollows is more
favorable for $N_2$ fixers (Leppanen et al., 2015; Zivkovic et al., 2022). By contrast, Vile et al. (2014) observed
higher BNF rates in the hummock species *S. fuscum* than in the hollows species *S. angustifolium.* All moss
samples for our $^{15}N$ assimilation experiment were collected in lawns. One exception was a subordinated number
of plants of *S. cuspidatum* typical of hollows in the BRU incubation. While the moss species were identical in
the UHL and MMJ incubation *(S. girgensohnii),* the BNF potential at these two sites was strikingly different
(Fig. 3). Therefore, we suggest that *Sphagnum* species was not a key BNF control in our $^{15}N_2$ experiment.

*4.3.5. Organic N availability*

Wang et al. (2022) stressed the positive effect of organic N on BNF. Assimilation cost of amino acids was
shown to be lower than that of $NH_4^+$ (Liu et al., 2013; Song et al., 2016). Organic N molecules can also serve as
a C source for cyanobacteria, thus saving the cost of photosynthesis (Krausfeld et al., 2019). As seen in Tab. S3,
concentrations of total organic N (TON) in bog water increased in the order: MMJ < BRU < UHL, and were thus
probably unrelated to augmented BNF at MMJ *sensu* Wang et al. (2022).

*4.3.6. Possible P limitation*



Phosphorus is needed for the synthesis of ATP playing a key role in symbiotic BNF (Rousk et al., 2017; Wieder
et al., 2022). In plant tissues, N:P ratios greater than 16 may indicate P limitation, while N:P ratios lower than 16
correspond to N limitation (Koerselman and Meuleman, 1996). Caution must be exercised in interpreting N:P
ratios in atmospheric deposition as potential controls of P or N limitation. In addition to atmospheric input
fluxes, bioavailable P and N in bog waters are strongly affected by a tight inner cycling with additional inputs
from biomass decomposition (Walbridge and Navaratnam, 2006). Phosphorus input fluxes *via* atmospheric
deposition into peat bogs may affect nutrient limitation in the long-run, depending on whether these input fluxes
are large enough, compared to the frequently observed P leaching to deeper peat layers (Walbridge and
Navaratnam, 2006, and references therein). According to Tab. S3, atmospheric deposition at all three study sites
is consistent with P limitation that might limit BNF (high N:P ratios of 169, 60, and 112 at BRU, UHL, and
MMJ, respectively). At the same time, N:P ratios in surface bog water were below 16 at two of the three sites,
UHL (7.6), and MMJ (15). At BRU (N:P = 24), P limitation inferred from bog water chemistry would provide an
explanation of non-detectable instantaneous BNF. At UHL, we found no indication of a relationship between P
availability and zero BNF. The relatively P-rich bog water (165-490 µg P L$^{-1}$; Tab. S3) at all sites may contain,
in addition to deposited P and mineralized P released during peat degradation, also, to some extent, geogenic P.
Bedrock granite (BRU, UHL) contains P in accessory apatite and K-feldspar whose weathering was probably
more efficient during the recent 40 years of acid rain. Phosphorus in phyllite (MMJ) is concentrated in apatite.
Phosphorus concentrations in fresh bedrock were similar at BRU and MMJ (52-55 ppb), and twice lower at UHL
(29 ppb; Gurtlerova et al., 1997; Pecina, 1999). The possible input of bioavailable geogenic P depended on local
hydrology and could be site-specific.
Living *Sphagnum* had N:P ratios of 31, 12, and 7 at BRU, UHL, and MMJ, respectively (Tab. S4), indicating
conditions favorable for BNF at the latter two sites. As seen in Fig. 7h, N:P < 16 marking N-limitation was
characteristic of the entire vertical peat profile at UHL, and downcore to a depth of 15 cm at MMJ. In contrast,
the N:P ratio was above 16 throughout the vertical peat profile at BRU. Phosphorus availability inferred from
bog water and living *Sphagnum* gave consistent results with respect to possible BNF. As mentioned above, P
likely limited BNF only at BRU.

Recently, measurements of regional P deposition started in headwater catchments of the GEOMON network
(Oulehle et al., 2017). In the time period 2014-2018, UHL, a site directly included in the GEOMON network,
exhibited lower P concentrations in the atmospheric input, compared to JEZ in the west (a proxy of BRU) and
UDL in the east (proxy of MMJ; *see* Fig. 1 for catchment locations; the distance between JEZ and UDL, and
between BRU and MMJ was approximately 70 km). Four-year average P concentrations at UHL were 72 and 36
µg L$^{-1}$ in open-area precipitation and spruce throughfall, respectively. At JEZ, analogous P concentrations were
103 and 87 µg L$^{-1}$, At UDL, these sample types contained on average 110 and 91 µg P L$^{-1}$. The high P uptake by
tree canopy resulting in low P concentration in throughfall might indicate P deficiency in UHL inputs. At the
same time, the N:P ratio in total vertical atmospheric deposition was lower than 16 at all three sites (13.1 at JEZ,
15.5 at UHL, and 13.7 at UDL (GEOMON Hydrochemical Database, Czech Geological Survey).

*4.3.7. Possible Mo limitation*



Nitrogenase requires molybdenum (Mo) in its active center to reduce $N_2$ to bioavailable $NH_4^+$ (Rousk et al.,
2017; Bellenger et al., 2020). In principle, Mo limitation of BNF may have played a role in the contrasting BNF
potentials observed at our sites. We do not have data on Mo concentrations in the studied ecosystems, except for
trace metal analysis of the prevailing rock types ($\leq 1$ ppm; Gurtlerova et al., 1997). However, known Mo
contents in coal massively mined/burnt in the Central European industrial region could shed some light on Mo
availability *via* atmospheric deposition: North Bohemian soft coal (Sokolov basin close to BRU; Fig. 1) contains
on average 18 ppm Mo, whereas Upper Silesian stone coal (Ostrava close to MMJ; Fig. 1) contains only ~0.6
ppm Mo, *i.e.,* 30 times less (Bouska et al., 1997). Since UHL is situated downwind of the North Bohemian
cluster of coal-burning power plants, and very close to Turow (soft coal mining in the Polish part of the Lusatian
basin; Fig. 1), atmospheric Mo inputs at UHL may be relatively high. Preliminarily, it appears to be unlikely that
Mo significantly influences the contrasting BNF potentials at our study sites.

*4.3.8. The role of $SO_4^{2-}$ deposition*

Large atmospheric inputs of acidifying sulfur forms ($SO_2$, $H_2SO_4$), characterizing northern Czech Republic since
the 1950s (Hunova et al., 2022), can affect BNF in two ways: by suppressing methanogenesis, and by reducing
the pH. Sulfate in peat bogs under high S deposition becomes an important electron acceptor (Pester et al., 2012)
and bacterial sulfate reduction is thermodynamically favored relative to methanogenesis and fermentative
processes (Vile et al., 2003). It not only decreases gross $CH_4$ production in peat, mitigating the flux of $CH_4$ to the
atmosphere and minimizing climate warming, but also lowers the supply of $CH_4$ to methanotrophs that, at some
sites, represent a major BNF pathway (Dise and Verry, 2001; Vile et al., 2014). Large $SO_4^{2-}$ inputs may thus
suppress BNF in peat bogs. In this context, is should also be mentioned that a $^{34}S/^{32}S$ isotope study has
documented post-depositional vertical mobility of S in industrially polluted peat bogs (Novak et al., 2009).
While long-term vertical S deposition, calculated according to Oulehle et al. (2016), was similarly high at UHL
and MMJ (means of 18.6 and 17.0 kg ha$^{-1}$ yr$^{-1}$ for the 1900-2012 period), higher than at BRU (12.2 kg ha$^{-1}$ yr$^{-1}$),
UHL bog water at the time of this study was nearly 70 times richer in $SO_4^{2-}$ than MMJ bog water, and eight
times richer in $SO_4^{2-}$ than BRU bog water (Tab. S3). Runoff at UHL was 4-5 times richer in $SO_4^{2-}$ than runoff at
MMJ and BRU. The zero instantaneous BNF at UHL in our $^{15}N_2$ incubation can be related to the highly elevated
S deposition in the case that UHL primarily hosts methane oxidizing diazotrophs.

UHL waters were characterized by lower pH, compared to those at MMJ and BRU (Tab. S3). Runoff pH at UHL
was 4.48, while runoff pH at MMJ was 7.40. Bog water pH at UHL was 4.02, while pH at MMJ was 4.88.
Downregulation of BNF in more acidic environment has been reported, *e.g.*, by Basilier (1979) and van den
Elzen et al. (2017). Accordingly, lack of BNF at UHL may be related to its lower pH, compared to the other two
study sites.

*4.4. Natural-abundance N isotope systematics*

*Sphagnum* metabolizes bioavailable $NH_4^+$ approximately eight times faster than $NO_3^-$ (Saiz et al., 2021). Because
there were nonsignificant differences between $\delta^{15}N$ values of $NH_4^+$ and $NO_3^{-\ 1}$ in rainfall at our study sites (Fig.





S1), it is reasonable to use the entire $\delta^{15}N$ data set for a comparison with $\delta^{15}N$ values of living *Sphagnum* (Fig.
4). Slow lateral mixing of surface bog waters may bring throughfall N from the forested margins of each bog to
the central unforested area and, therefore, we additionally included throughfall $\delta^{15}N$ data in Fig. 4 comparisons.
The isotopically analyzed living *Sphagnum* plants represented on average a one-to-two-year increment (*cf.*,
Wieder and Vitt, 2006). We found a statistically significant shift from isotopically light N of the deposition to
isotopically heavier N of *Sphagnum* only at BRU ($p < 0.05$). This might indicate mixing with even heavier
atmospheric $N_2$ taken up by diazotrophs. At BRU, BNF might have intermittently proceeded over the most
recent growing seasons even though the $^{15}N_2$ experiment did not corroborate this process in October 2018.

A straightforward attribution of the N isotope pattern at BRU to BNF, however, is hampered by the fact that
mineralization is a likely alternate source of dissolved $N_r$ for assimilation by the moss (Zivkovic et al., 2022, and
references therein). The often found high $\delta^{15}N$ values of mineralized $N_r$ remaining in the bog ecosystem result
from an isotope fractionation accompanying denitrification, a process known to occur especially in peat bogs
that are not extremely acidic. Gaseous products of denitrification contain isotopically light N both in wetlands
(Denk et al., 2017; for data from Czech peat bogs *see* Novak and al., 2015a, 2018), and aerated forest soils
(Houlton and Bai, 2009; for data from Czech upland soils *see* Oulehle et al. 2021a). Nitrogen in surface bog
water at BRU had a positive mean $\delta^{15}N$ value of 0.9 ‰ (Fig. 6). Isotope systematics at BRU are thus consistent
with incorporation of mineralized $N_r$ into moss biomass during assimilation instead of uptake of N resulting from
BNF.

Advancing mineralization accompanying peat maturation with mobilization and export of gaseous low-$\delta^{15}N$
nitrogen is also responsible for the increasing $\delta^{15}N$ values of the residual peat substrate downcore (Fig. 7d).

Fig. S3 summarizes two general scenarios, under which a difference between N isotope composition of
atmospheric input, *Sphagnum* and bog water indicates BNF: (1) the mean $\delta^{15}N$ values increase in the order:
deposited $N_r$ < bog water $N_r$ < *Sphagnum* $N_r$ < atmospheric $N_2$, or (2) the mean $\delta^{15}N$ values decrease in the
order: deposited $N_r$ > bog water $N_r$ > *Sphagnum* $N_r$ > atmospheric $N_2$. Whereas the $\delta^{15}N$ value of bulk
atmospheric deposition in Central Europe is mostly negative, positive mean $\delta^{15}N$ values have been reported from
other regions. One example is isotopically heavy N of dry-deposited $HNO_3$ in an industrial part of the U.S.
(Elliott et al., 2009). Fig. S3 assumes that the magnitude of potential N isotope fractionations during uptake of
inorganic N into plant biomass is relatively small and does not overprint the larger N isotope differences
between the above discussed mixing endmembers.

It remains to be seen how to reconcile the relatively high instantaneous BNF rate at MMJ, measured in the
laboratory, with the non-existence of a positive $\delta^{15}N$ shift from atmospheric deposition (mean of -4.3 ‰) to
*Sphagnum* (mean of -4.4 ‰; Fig. 4; $p > 0.05$). Given that we explained the positive $\delta^{15}N$ shift from deposition to
*Sphagnum* at BRU by mixing of low-$\delta^{15}N$ rainfall with high-$\delta^{15}N$ bog water, and that bog-water N at MMJ is
isotopically heavy, a similar positive N isotope shift from rainfall to *Sphagnum* would be expected also at MMJ.
Such was not the case. This observation is important because it might indicate that uptake of recently
mineralized $N_r$ from bog water at sites hydrologically similar to MMJ (and also BRU) may not control the N




isotope signature of living *Sphagnum*. An input of isotopically light $N_r$ for assimilation by the MMJ moss could,
in principle, originate from shallow groundwater upwelling or lateral water inflow from other segments of the
catchment possibly bringing legacy low-$\delta^{15}N$ nitrogen from the peak acid-rain period throughfall. Such within-
site water inputs could affect the intermediate $\delta^{15}N$ value of *Sphagnum* at MMJ.

**Conclusions**

Based on hydrochemical monitoring data and statistical modelling, the three studied *Sphagnum* peat bogs located
in the industrial northern Czech Republic received close to 18 kg $N_r$ ha$^{-1}$ yr$^{-1}$ *via* atmospheric deposition. Since
1900, the atmospheric input of $N_r$ affected the study sites in the order: UHL > MMJ > BRU. In the most recent
years, the annual $N_r$ inputs *via* vertical deposition between the sites differed by mere 1 to 2 kg ha$^{-1}$ yr$^{-1}$. The sites
can thus be classified as highly to medium-polluted. A 168-hour $^{15}N_2$ assimilation experiment revealed relatively
high but variable rates of BNF at MMJ, and non-detectable BNF at the remaining two sites, characterized by
slightly higher and slightly lower $N_r$ depositions, respectively, compared to MMJ. We investigated in all 10
different parameters that might have served as controls of the presence or absence of instantaneous BNF in
living moss. In addition to bulk $N_r$ deposition fluxes, these parameters included: $NH_4^+$-N/$NO_3^-$-N ratio in
atmospheric input, temperature, wetness, *Sphagnum* species, organic-N availability, possible P limitation,
possible Mo limitation, $SO_4^{2-}$ deposition, and pH. Using the available data, we argue that P deficiency was the
likely inhibitor of BNF at BRU. Assuming that methanotrophic bacteria represented a major type of diazotrophs,
extremely high $SO_4^{2-}$ inputs may have been the key control of the absence of BNF at UHL. While the long-term
temperature and wetness at UHL were also lower, compared to the remaining two sites, they probably did not
affect the results of the $^{15}N_2$ experiment since the incubation was performed under the same temperature and
wetness for all sites. In general, higher concentrations of decomposition-inhibiting metabolites could be causally
related to BNF rates. Such a control of BNF was unlikely since the same *Sphagnum* species from MMJ and UHL
was used for the $^{15}N_2$ experiment that showed contrasting results for these two sites. Tthe large $\delta^{15}N$ differences
between moss replicates that were collected from various segments of MMJ at the end of the $^{15}N_2$ incubation
suggested an existence of BNF hotspots.

The use of natural-abundance N isotope ratios to corroborate the observed instantaneous BNF rates was
hampered by isotopically heavy N of surface bog water. The bog water contained secondary $N_r$ forms which
could have resulted from partial *Sphagnum*/peat decomposition and removal of the complementary low-$\delta^{15}N$
products of denitrification. At BRU, we found statistically significant differences in $\delta^{15}N$ values in the order:
deposited $N_r$ < *Sphagnum* $N_r$ < atmospheric $N_2$ < bog water $N_r$. Stable isotope ratios could not unambiguously
distinguish between assimilation of bog-water $N_r$ and atmospheric $N_2$ to form the observed N-isotope signature
of *Sphagnum*. At UHL and MMJ, $\delta^{15}N$ differences between *Sphagnum* and the atmospheric input were
statistically insignificant. The natural-abundance approach as a test of BNF presence may give more promising
results at high-latitude sites often characterized by greater (30-40 cm) depth of the water table level below
*Sphagnum* capitula than the Central European sites.



**Author contribution:** M. Stepanova: conceptualization, data curation, visualization, writing – review and
editing; M. Novak: conceptualization, data interpretation, writing – original draft; B. Cejkova: methodology,
nitrogen fixation data acquisition, data interpretation; I. Jackova: methodology, concentration and isotope data
acquisition; F. Buzek: methodology, data interpretation, validation; F. Veselovsky: field work; J. Curik: field
work; E. Prechova: formal analysis, resources; A. Komarek: statistical analysis; L. Bohdalkova: data
interpretation

**Competing interests.** The authors declare that they have no conflict of interest.

**Acknowledgements.** This is a contribution to the Strategic Research Plan of the Czech Geological Survey
(DKRVO/CGS 2018-2022, grant. no. 310660 to MS). We thank Prof. Martin Sanda of the Czech Polytechnic,
Prague, and Jan Knotek of the Jeseniky branch of the Czech Geological Survey for field work assistance. Dr.
Filip Oulehle is thanked for modelling of long-term atmospheric N deposition at the study sites, and Oldrich
Myska for providing monitoring data from the GEOMON database.

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
