# Peer review of "Contrasting potential for biological N2-fixation at three"

_EGUsphere, 2023_

## Author Comment (AC1)

**Anonymous Referee #1 (*text in italics*):**

*Overall comments:*
*The manuscript compares the biological nitrogen fixation (BNF) potential at three polluted peat bogs of central Europe. The topic is of great interest taking into account the key role of BNF in the availability of N in peatlands. These ecosystems could be a source of greenhouse gases, especially disturbed polluted ones, and very few studies have been done in historically highly polluted areas. In addition, they also provide insight into biotic and abiotic controls over BNF.*
*The manuscript fits well with the SOIL aims and scope. In general, it is very well organised and presented. However, there are several causes of major concern that prevent accepting the paper for publication. In the first instance, from the general perspective, the authors need to build a stronger case to convince the reader of the results obtained from one single BNF measurement in time. Other parameters have been measured over years, but BNF at each site is just one single measurement, and not in-situ but in the laboratory.*

**Our response:** We present replicated (n = 4) delta15N measurements at time t = 0 and t = 7 days for three sites. In the revised version, we are ready to add similar measurements at t = 2 days (three sites, n = 2) which we performed as a preliminary trial (it gave data consistent with the 7-day experiment). The total of delta15N measurements in the BNF incubation experiment at the Czech sites is 42. The total number of delta15N measurements in our entire paper is 403. It was our intention to perform more natural-abundance N isotope measurements as part of the in-situ monitoring at 3 study sites than delta15N measurements as part of the Sphagnum incubation study (361 vs. 42). In the moss incubation study, we mainly wanted to illustrate extremely different BNF rates among sites and within sites with similar N pollution (these data are in Fig. 3). At the same time, we intentionally paid a lot of attention to the other part of the project, the delta15N monitoring under natural conditions. That type of research appeared to be an underexploited approach to assessing BNF occurrence. The overarching motivation for our study design was to present simultaneously short-term BNF results (15N2 assays) and longer-term results (natural-abundance N isotope monitoring). Ideally, the results would be complementary. What the reviewer views as a potential weakness (single BNF measurements vs. parameters measured over years), was intended by us to be a strength. Please note that some of the landmark papers in this field offer lower or similar number delta15N incubation measurements relative to the current study. For example, Vile et al. 2014, performed 16 N isotope measurements, van den Elzen et al., 2017, made 16 N isotope measurements, and van den Elzen et al., 2020, made 58 N isotope measurements. Our total number of delta15N measurements (403) is higher compared to the N isotope data set presented by Knorr et al., 2015 (72), Saiz et al., 2019 (100), and Saiz et al. 2021 (156). Our entire delta15N data set is somewhat lower than the N isotope dataset by Stuart et al., 2021 (580) and Zivkovic et al., 2022 (536). We understand the Reviewer's point that a time-series of incubation experiments using samples collected in different seasons would be needed. We have recently submitted a research proposal planning to do exactly that at newly selected sites. In our incubation, we used summer-time temperatures (June 21 – September 23) to show the potential BNF rates in the mid-growing season which we believed would be high, compared to spring and autumn. (BNF peak in summer was reported, e.g., by Zivkovic et al., 2022.) We are ready in include information on how we chose the incubation conditions in light of meteorological data from the studied areas.

*In addition, some more specific questions regarding the methodology must be clearly addressed and justified:*

*The authors indicate that surface bog water was collected in June 2019 at each study site (lines 175-176). And that Sphagnum mosses and peat were collected in October 2018 (line 180). How can be compared their δ15N value (e.g. lines 322-324) with sampling dates eight months apart?*

**Our response**: The surface bog water was used as part of the natural-abundance N-15 study, the data are reported in Fig. 6. These samples were collected in June 2019, as mentioned on lines 175-176. In contrast, collection in October 2018 is mentioned in association with the N-15 labelling study. For the N-15 labelling study, the used bog water was collected on the same day as Sphagnum in October 2018. Regarding the comparison between bog water and moss on lines 322-324: As seen in Fig. 6, the range of delta15N values of living Sphagnum was rather narrow (within 2.5 per mil). The homogenized green Sphagnum plants were 5 cm long, the growth increment represents probably more than 3 years, i.e., more than the 8-month interval between bog water and Sphagnum sampling. We know the typical increments from many 210 Pb dated profiles from the same Central European region. (We have published lead-210 dating of vertical Sphagnum peat profiles from the following 14 peat bogs in the area: Bozi Dar, Rybarenska slat, Bila Smeda, Tajga, Pancavska louka, Cervene blato, Pod Jeleni horou, Mrtvy luh, MMJ, Ocean, Velke jerabi jezero, Blatenska slat, Pod Zielencem, and Velke Darko; cf., Novak et al., 2003, Environ. Sci. Technol. 37:437; Novak et al., 2008 STOTEN 390:426). At the depth of 4 cm, peat is on average 3.3 years old; at the depth of 6 cm, the peat is on average 7.2 years old. Even if we performed several samplings of bog water, it probably would not cover the entire time span of the sampled living Sphagnum plants. We suggest to delete the statement on lines 322-323 and/or to add references to 210-Pb dates of living Sphagnum in similar environments.

*Related to the above (may answer it), in Table S3, "Data from October 2018" is for all the data provided by the table? It does not add up with the information provided in the materials and methods section as mentioned before nor in Table S2. It should be clear that the comparisons are among samples collected on the same date, or otherwise justify why they are comparable.*

**Our response**: We did not analyze the entire chemism (17 variables) in all monitoring samples presented in Tab. S2. The chemism in Tab. S3 was intended to illustrate typical local conditions. All these values were measured in samples collected within the same month, i.e., in October 2018. In a revised version, we can stress the snapshot character of info in Tab. S3.

*The authors mention that they collected live Sphagnum and transported it to the laboratory at 6 °C (lines 182-186) and later on they talk about the incubation experiment (lines 190-214). However, several questions arise:*
*How long it took from the collection in the field to the laboratory? And to the incubation?*

**Our response:** The journey from the field sites to the laboratory took 2 to 4 hours. The time elapsed between moss collection and the start of the laboratory incubation was as follows: 2 days in the case of BRU, 3 days in the case of UHL, and 6 days in the case of MMJ. Due to the geographical distribution of the 3 study sites, it was not feasible to collect moss samples on a single day for the concurrent incubations.

*How were the live Sphagnum samples maintained in the laboratory?*

**Our response:** Prior to the start of the incubation experiment, the Sphagnum samples were kept in a growth chamber at 6 °C. They came from the field saturated with bog water. Therefore, we did not moisturize them during storage.

*Was there an acclimatization period before the incubation?*

**Our response:** Yes, there was an acclimatization period before the beginning of the experiment: 4 hours at room temperature of 22 °C.

*What may be the shortcomings (or reasons) of laboratory incubations instead of in-field ones? Have these been considered?*

**Our response:** According to Myrold et al. (1999), "because of the need for gas-tight assay system, 15N2 incorporation is better suited for laboratory than field studies." A survey of the recent literature indicates that the following major papers on BNF rates in peat also preferred a laboratory experiment under controlled conditions: Stuart et al. 2021, Rousk et al., 2018, Warren et al., 2017, van den Elzen et al., 2017, and Knorr et al., 2015. On the other hand, in-situ incubations were performed, for example, by Zivkovic et al., 2022, Saiz et al., 2021, van den Elzen et al., 2020, Rousk et al., 2018, and Vile et al., 2014.

*The authors explain that their laboratory conditions during the incubation period were 16 h day at 18 °C and 8 h night at 10 ° However, they do not justify the reason. Is this setup mimicking real conditions at the time of sampling? Is it just optimal conditions for the BNF process? It needs to be justified and put in context.*

**Our response:** The chosen incubation temperatures/light duration did not mimic real conditions at the time of sampling. We used an approximation of summer-time temperatures (18 °C day, 10 °C night) at the study sites. Since continuous temperature measurements are not performed directly at the study sites, we used temperature data from nearby meteorological stations operated by the Czech Hydrometeorological Institute. Throughout the Czech Republic, this institute runs a total of 217 stations, many are situated at elevations significantly lower than those of our mountain-top peat bogs. Our temperature approximation was based on data from similar altitudes and stations typically located less than 20 km from the study sites. We view our BNF estimates as potential rates during the three summer months (June 21-September 23). Similar "optimal" incubation conditions were used by van den Elzen et al., 2017 (18 °C, 16 hours of light per day).

*Line 196. Here it is indicated that the incubation lasted 168 hours, which is 7 days. The authors noted (lines 207-210) that this is a longer incubation than most previous studies, but do not explain the reason. This incubation time should be justified. Here the authors should address what potential errors or shortcomings are associated with such a long incubation, aside from changing headspace concentrations of gases. This is important to explain clearly because literature suggests for this type of BNF measurements, short-term incubations, i.e., less than 4 days (Myrold et al., 1999).*

**Our response:** Before measuring delta15 values in a 7-day incubation, we tested whether a delta15N shift toward higher values would be measurable after 2 days (Myrold et al., 1999 recommend experiment duration of less than 4 hours). In the 2-day experiment, we arrived at the following del15N values (in per mil): MMJ: t = 0 …-3.7 and -3.7, t = 2 days … +2.7 and +3.3; BRU: t = 0 …-4.0 and -4.0, t = 2 days … -3.8 and -3.4; UHL: t = 0 …-4.0 and -4.0, t =

2 days … -3.9 and -3.5. Two delta15N values differing by less than 0.3 per mil are indistinguishable. We perform the longer 7-day l incubation to obtain more distinct trends toward higher delta15N of the moss at the end of the experiment, compared to the 2-day data. We minimized the effect of changing gas chemistry in the sealed headspace toward the end of a longer experiment by using relatively large containers. Specifically, while most experiments in the literature use 50 to 60 mL containers, we used air-tight 200 mL containers. We were able to show that even in days 3 to 7, the delta15N values at MMJ continued to grow. At BRU and UHL, BNF was triggered off neither within the 2-day, nor 7-day period. We are ready to present these data in the new version of the manuscript.

*Minor comments:*
*Line 89: delete the comma after "Zivkovic et al."*

**Our response:** O.K. Thank you.

*Line 227: It is mentioned an "Appendix I". I was not able to find it. Was it provided?*

**Our response:** Appendix I is part of the submission now.

*Line 310: "MMJ mean of 1.0 wt." should say "1.1 wt."*

**Our response:** O.K. Thank you.

---

## Author Comment (AC2)

**Appendix I**

**Methodology of chemical analysis of liquid samples**

Concentrations of $Na^+$, $K^+$, $Ca^{2+}$, $Mg^{2+}$, $Mn^{2+}$, and $Fe_{tot}$ were measured by flame atomic absorption spectrometry (FAAS; AAnalyst 100, PerkinElmer) with the limits of quantification (LOQ) of 0.01 mg $L^{-1}$ and 0.005 mg $L^{-1}$ for Fe. Concentrations of $NH_4^+$ and $P_{tot}$ were determined spectrophotometrically (PMT; Perkin-Elmer Lambda 25; LOQ of 0.02 and 0.006 mg $L^{-1}$, respectively). Concentrations of $Cl^-$, $SO_4^{2-}$ and $NO_3^-$ were determined by ion chromatography (HPLC; Knauer 1000; LOQ of 0.15, 0.5, and 0.3 mg $L^{-1}$, respectively). Concentrations of $F^-$ were measured potentiometrically (ION 85 Radiometer Inc.; 0.02 mg $L^{-1}$). Concentrations of $HCO_3^-$ were measured by titration (LOQ of 0.6 mg $L^{-1}$). Dissolved organic carbon (DOC) and total dissolved nitrogen (TN) were determined on an Apollo 9000 analyzer (Tekmar-Dohrmann; LOQ of 0.1 and 0.5 mg $L^{-1}$). Measurement of pH was carried out on PHM-62 Radiometer, and conductivity on CDM-83 Radiometer Denmark.

**Methodology of chemical analysis of solid samples**

Ash content in peat was determined on a 0.5 g aliquot at 550 °C. Concentrations of Na, Mg, K, and Ca were measured by flame atomic absorption spectrometry (FAAS; AAnalyst 100, PerkinElmer) with the limits of quantification (LOQ) of 50 ppm. Phosphorus content was determined spectrophotometrically (P-E Hitachi 200; LOQ of 50 ppm). A 10-mg aliquot of each homogenized peat sample was placed in a tin capsule and combusted in a Fisons 1108 elemental analyzer at 1040 °C. Carbon concentrations in peat were determined with a reproducibility of 1.0 %.

---

## Author Comment (AC3)

**Anonymous Referee #2:**

*The biological fixation of nitrogen (or BNF) is a very important soil process yet is plagued by large spatial and temporal variability, arising from the large numbers of variables (environmental, biogeochemical and microbial) which can influence the magnitude of the process. It is particularly important in soil systems that have a limited alternative source of nitrogen or those that have been affected by pollution. One such system is peatlands, which are supplied primarily by precipitation, resulting in generally ombrotrophic conditions, and which have also been affected by atmospheric deposition of pollutants such as N and S compounds.*

*This manuscript is a useful addition to the literature on being able to bring together possible explanations for the variations in the rates of BNF and the manuscript contains an extensive review of the literature which addresses this topic. The primary contribution is to show that three central European peatlands at a high elevation receiving substantial atmospheric deposition of N and containing Sphagnum moss have very different rates of BNF and the study seeks to find why, using two main approaches. One is incubation of Sphagnum moss samples with labeled 15N2 and the second is to use natural abundance variations in the 15N isotope composition of the plant material, water and precipitation. The 'usual suspects' controlling BNF are examined with the measurements available, or deduced from alternative sources.*

*The main conclusion is that one site appears to be affected by a paucity of P and one by a high concentration of SO4, resulting in essentially no BNF, with the third site showing the largest rate of BNF, but without any clear indicator of why, though the weaker knowledge of hydrology at the site may be a factor. The occurrence of methanotrophic bacteria as a component of BNF requires evidence that methane is available in the location where oxidation will occur and incubation of samples with ambient methane concentration is unlikely to identify that source. The laboratory conditions for the BNF assessment were somewhat unusual and 'one-time', whereas there are likely substantial variations in field conditions. There is a suggestion that at the BNF-active site, microbes may have adapted to the high atmospheric N loading (from another paper), though it was the same at the other sites.*

*The natural abundance assessment is complicated because of all the changes in 15N that may be brought about by N transformations, and the presence of N uptake by Sphagnum from N in peat water produced by the mineralization of the peat and litter, and these uncertainties are recognized. On top of this, the 15N sampling at the site with substantial BNF showed a large spatial variability which suggested small-scale variations in BNF, or 'hot spots' and possibly 'hot moments'. One question occurred to me: Sphagnum N concentration was larger at the active-BNF site than the other two (Fig. 5) but the underlying peat (0-10 cm) had a smaller N concentration (Fig. 7). Any reason for that change?*

**Our response:** Please note the small letters in the superscript in Fig. 5. BRU and MMJ are marked with "b". That means that these two sites are statistically indistinguishable in terms of N concentrations in Sphagnum. We cannot say that N concentration was larger at the BNF-active site than at the other two. We are reluctant to conclude that the underlying peat had a smaller N concentration at MMJ than at the other two sites at the wetland scale because we had only one vertical profile in Fig. 7 and as many as 21 Sphagnum analyses in Fig. 5. For such a comparison, we would need a number of vertical peat profiles at each site. Please note that the topmost peat sample in Fig. 7 is essentially living Sphagnum. The one sample at MMJ had low [N] in Fig. 7 while Fig. 5 shows a surprisingly large [N] variability at MMJ,

compared to the other sites. In our view, the top low-[N] sample in the MMJ peat core only confirms the large [N] variability in Fig. 5 bottom.

*I found the manuscript to be well structured, written and illustrated with a substantial linking to previous studies. It is 'interdisciplinary' (as much as 'disciplines' still exist), drawing upon atmospheric, biogeochemical and biological controls on the BNF process in an edaphic context. On the whole, though, some of the results are inconclusive because of a lack of measurements to assess all the variables that may affect BNF, but that is the nature of the topic undertaken. I found a few typographical errors, which should be readily correctible.*

---

## Author Response (AR2)

To: Kate Buckeridge

Dear Kate,

Attached you will find the two "corrected" graphs for the Electronic Annex of our paper. We have double-checked, the files we originally uploaded to the journal's electronic system were correct. I.e., they did contain deltas, not gammas. So we are uploading the same graphs again - we cannot do anything else. The problem may originate from your opening of the graph files. If there is anything we can do, please let us know.

Thank you very much for your patience.

Sincerely,

Martin Novak
Corresponding author